# Training DNNs Resilient to Adversarial and Random Bit-Flips by Learning Quantization Ranges

**Kamran Chitsaz** *kamran.chitsaz-zade-allaf@polymtl.ca*
*Polytechnique Montreal*

**Gonçalo Mordido** *goncalo-filipe.torcato-mordido@mila.quebec*
*Mila, Polytechnique Montreal*

**Jean-Pierre David** *jean-pierre.david@polymtl.ca*
*Polytechnique Montreal*

**François Leduc-Primeau** *francois.leduc-primeau@polymtl.ca*
*Polytechnique Montreal*

**Reviewed on OpenReview:** *https://openreview.net/forum?id=BxjHMPwZIH&*

## Abstract

Promoting robustness in deep neural networks (DNNs) is crucial for their reliable deployment in uncertain environments, such as low-power settings or in the presence of adversarial attacks. In particular, bit-flip weight perturbations in quantized networks can significantly degrade performance, underscoring the need to improve DNN resilience. In this paper, we introduce a training mechanism to learn the quantization range of different DNN layers to enhance DNN robustness against bit-flip errors on the model parameters. The proposed approach, called weight clipping-aware training (WCAT), minimizes the quantization range while preserving performance, striking a balance between the two. Our experimental results on different models and datasets showcase that DNNs trained with WCAT can tolerate a high amount of noise while keeping the accuracy close to the baseline model. Moreover, we show that our method significantly enhances DNN robustness against adversarial bit-flip attacks. Finally, when considering the energy-reliability trade-off inherent in on-chip SRAM memories, we observe that WCAT consistently improves the Pareto frontier of test accuracy and energy consumption across diverse models [1].

## 1 Introduction

The widespread application of DNNs in critical domains, such as self-driving cars (Bose et al., 2021) and medical diagnosis (Gongye et al., 2020), has emphasized the need to study their reliability and performance in noisy environments. Also, to improve their performance, custom hardware accelerators have emerged as a promising solution given the high computational demands of DNNs. However, it is essential to ensure that DNNs retain their efficiency and reliability when deployed on specialized hardware with a limited energy budget and in the presence of hardware faults. Although quantizing and compressing DNN weights can improve robustness and decrease the potential impact of bit-flip errors (Qian et al., 2023), adversarial attacks can still cause severe performance deterioration with only a few vulnerable bits (Rakin et al., 2019).

Bit-flips in memories can occur randomly as a result of undesirable events such as chip fabrication variations, supply voltage variations, or targeted attacks. The row-hammer attack (RHA), where multiple accesses can disturb DRAM rows and cause bit-flips in other rows, has been extensively researched (Kim et al., 2014), with DRAMs being highly susceptible to this type of attack due to their dense architecture. Furthermore,

---

[1]Code is available at https://github.com/kmchiti/WCAT

to improve energy efficiency in recent DNN accelerators, reducing the memory supply voltage has become a popular approach to reduce energy consumption, particularly in SRAMs, due to the quadratic relationship between dynamic energy consumption and supply voltage. Despite energy-efficient, near-threshold voltage processing can cause bit-flip errors in SRAMs, which may compromise the reliability of the DNN accelerator (Frustaci et al., 2015).

In recent years, there has been an increased focus on developing defense methods to improve the robustness of DNNs against random weight perturbations as well as adversarial attacks. The fault tolerance capability of DNNs can be improved by modifying the network's size, structure, and training process. For example, some methods improve the DNN robustness by adding memory or computational overhead (Liu et al., 2023; Guo et al., 2021), while others involve compressing and binarizing the weights of the DNNs to limit the impact of weight perturbations (He et al., 2020). However, this can lead to a performance loss when the model weights are not perturbed, which we refer to as the noiseless weights setting.

In the conventional noiseless setting, the quantization range creates a delicate trade-off between the quantization error and the dynamic weight range. While reducing the quantization range decreases the quantization error due to rounding, it also decreases the dynamic range, which is vital in neural networks for representing weights with large absolute values (Choi et al., 2019). Our experimental results demonstrate that the quantization range also plays a crucial role in network robustness. Hence, the goal of this work is to minimize the quantization range, while striking a good balance between robustness to noisy weights and performance on noiseless weights. By incorporating this learning mechanism during training, we manage to uphold DNN performance under noiseless conditions while achieving a more robust network against bit-flip errors at inference time.

Our proposed approach, weight-clipping aware training (WCAT), focuses on learning and reducing the quantization range of different layers, thereby enhancing robustness against bit errors during inference. To ensure efficient training convergence without compromising the performance of the noiseless network, we employ an accurate gradient approximation technique. Additionally, we simulate bit-flip errors during the training phase to further improve DNN robustness, leading to a more resilient model against bit errors during inference. Moreover, in the context of fine-tuning pre-trained DNNs for increased robustness, we demonstrate that initializing the quantization range based on the expected mean square error between real weights and dequantized noisy weights leads to higher robustness. Overall, our findings and results pave the way to advancing the understanding and practical implementations of robust DNN quantization methods.

## 2 Related work

This section begins by providing an overview of quantization techniques during training (Section 2.1). We then delve into adversarial attacks in Section 2.2. Finally, we review existing defense mechanisms against bit-flip attacks or perturbations in Section 2.3.

### 2.1 Quantization-Aware Training

Low-precision neural networks can benefit from quantization operations during the training process to learn quantization-aware parameters and improve performance. Although the quantization range is usually set to the maximum absolute value of the weight to ensure that the largest integer represents the largest weight value within the bit precision, it may cause high rounding error and consequently low robustness. Finding a better quantization range can be achieved based on the statistics of the weights (Choi et al., 2019) or by duplicating channels with weight outliers and halving their values to avoid outlier distortion when performing weight clipping (Zhao et al., 2019).

Meanwhile, learning the quantization range during training has emerged as a promising method for low-precision quantization. However, this approach can be difficult because the quantizer itself is not continuous, so approximating the gradient of the quantizer is required. Among these methods, (Jung et al., 2019) proposed a clipping function that balances the trade-off between the quantization range and sparsification. Baskin et al. (2021) introduced a parameterized clipping function that is applied before quantization. Jain et al. (2020) proposed a quantization scheme with a learnable scaling factor and defined the gradient based

on the straight-through estimator (STE). Also, (Esser et al., 2019) used a similar method but added gradient scaling to achieve state-of-the-art performance on the ImageNet dataset.

## 2.2 Bit-Flip Attack

The bit-flip attack is a malicious technique that manipulates the parameters of a DNN model at runtime to reduce its performance. Although random bit-flips applied to a network's weights can corrupt performance, they can generally withstand a relatively large number of such flips before performance is significantly degraded. Therefore, an attacker is looking to identify vulnerable bits by leveraging information about the network architecture, the parameters, or the dataset. Rakin et al. (2019) propose a progressive bit search based on the approximated gradient with respect to the bits of the network. The goal of this method is to reduce the network's performance to a random guess. The objective function of the attack is given by

$$
\begin{aligned}
&\max_{\{\hat{\mathbf{B}}_l\}} \mathcal{L}\left(f_{\{\hat{\mathbf{B}}_l\}_{l=1}^{L}}(X), Y\right) \\
&\text{s.t.} \sum_{l=1}^{L} \mathcal{H}\left(\hat{\mathbf{B}}_l, \mathbf{B}_l\right) \le N_b,
\end{aligned}
\tag{1}
$$

where $\mathcal{L}$ is the loss function, $\mathbf{B}_l$ is the binary representation of the weights at layer $l$, $\hat{\mathbf{B}}_l$ is the perturbed binary representation of the weights attacked by the bit-flip attack (BFA), $X$ is a training input, and $Y$ its corresponding label. The constraint in equation 1 is given in terms of the Hamming distance $\mathcal{H}$ between the attacked binary weights and the original binary weights, and $N_b$ is the maximum number of bit-flips allowed.

## 2.3 Defense Methods

Various defense techniques have been proposed to enhance the robustness of DNNs against bit-flip attacks or perturbations. Some methods involve additional overhead, such as using error-correction code units within SRAM cells (Jahinuzzaman et al., 2009) or implementing hash-based weight verification schemes to correct errors before inputting data into the DNN (Guo et al., 2021). Others, like binarization (He et al., 2020) and adversarial training (Buldu et al., 2022), can increase robustness at the expense of sacrificing the performance of the noiseless model. Additionally, Rakin et al. (2021b) proposed augmenting the model's capacity during training, which further enhances robustness, in conjunction with binarization.

In a recent study, Stutz et al. (2021) proposed an innovative approach to improve the robustness of neural networks by combining asymmetric fixed-point quantization with aggressive weight clipping during training. In addition, Stutz et al. (2022) also proposed to apply per-layer weight clipping, where a specific quantization range is set for each layer. This approach provides additional robustness, as different layers may require different quantization ranges depending on the magnitude and distribution of the weights. By applying per-layer weight clipping, the network can better adapt to these differences and optimize the quantization ranges for each layer accordingly. In our approach, we also leverage weight clipping, but with a key distinction: instead of predefined ranges, we focus on learning and dynamically reducing the clipping range throughout training. This dynamic adaptation further enhances network robustness. Importantly, our paper empirically demonstrates the pivotal role of quantization range in mitigating bit-flip noise, an aspect not extensively explored in prior research. Furthermore, our motivation for learning the quantization range through optimization is grounded in empirical evidence showing significant improvements in the performance of quantized models (Nagel et al., 2021).

## 3 Weight Clipping-Aware Training (WCAT)

In our proposed approach, Weight Clipping-Aware Training (WCAT), we draw inspiration from our empirical analysis on the optimal quantization range in the presence of bit-flip errors (see Appendix A). Our focus lies on uniform symmetric signed quantization for the weights of both convolutional and fully-connected layers. To reduce the quantization error, we first clip the weights and then apply linear weight quantization

(Section 3.1). Then, we employ the STE (Bengio et al., 2013) mechanism to update the weights that are both inside and outside the clipping range during training (Section 3.3).

Additionally, the optimal clamping value for each layer is also learned (Section 3.2) and is further reduced during training using weight decay. Moreover, we discuss optionally applying bit-flip noise injection during training (Section 3.4) to further improve the network robustness in highly noisy regimes. We summarize our proposed approach in Section 3.6.

### 3.1 Weight clipping and quantization

Although asymmetric quantization has shown better robustness against bit-flip noise (Stutz et al., 2021), symmetric quantization has the advantage of reducing the computational overhead of handling the zero-point offset during accumulation operations (Nagel et al., 2021) and is more hardware-friendly. Hence, we adopt uniform symmetric quantization and use the following clipping function that constrains the weight values of layer $l$, $W_l$, to be in a symmetric range controlled by $\alpha_l \in \mathbb{R}^+$:

$$\bar{W}_l = \text{Clip}(W_l, \alpha_l) = \begin{cases} -\alpha_l, & W_l < -\alpha_l, \\ W_l, & -\alpha_l \leq W_l \leq \alpha_l, \\ \alpha_l, & W_l > \alpha_l, \end{cases} \tag{2}$$

where $\bar{W}_l$ represents the clipped weights in the range $[-\alpha_l, \alpha_l]$. We perform such clipping in a layer-wise manner, particularly in every convolutional or fully-connected layer of the network. After clipping, we then map the real-valued weights to a discrete grid of integers using a linear quantization scheme:

$$W_l^{\text{int}} = \left\lfloor \frac{2^{b-1} - 1}{\alpha_l} \times \bar{W}_l \right\rceil, \tag{3}$$

where $b$ is the bit precision and $\lfloor \cdot \rceil$ represents the round-to-nearest integer operator. To approximately recover the real-valued weights $\bar{W}_l$ from their quantized representations $W_l^{\text{int}}$, we apply the following dequantization step:

$$\hat{W}_l = \frac{\alpha_l}{2^{b-1} - 1} \times W_l^{\text{int}}. \tag{4}$$

The intuition behind simply applying linear quantization rather than more complex quantization schemes, *e.g.* Mordido et al. (2019); Goncharenko et al. (2019); Esser et al. (2019), is that we hypothesize that the quantization range is what most matters in terms of robustness, as recently indicated by previous work and also observed in our experimental results. Nevertheless, we note that using more complex schemes may result in a better overall performance of the quantized, noiseless network.

### 3.2 Learned quantization range

To strike a better balance between reducing quantization error and preserving the dynamic range, thereby achieving enhanced robustness, we optimize the parameter $\alpha_l$ during training. The gradient of $\alpha_l$ w.r.t. each value of weights in layer $l$ is defined as follows:

$$\frac{\partial \hat{w}}{\partial \alpha_l} := \begin{cases} -1, & w_l < -\alpha_l, \\ 0, & -\alpha_l \leq w_l \leq \alpha_l, \\ 1, & w_l > \alpha_l, \end{cases} \tag{5}$$

with a different $\alpha_l$ being learned for each layer. Importantly, since the gradients from the weights outside the quantization range get accumulated, we introduce a scaling factor applied to the gradient of the quantization range $\alpha_l$ during the learning process, similar to Esser et al. (2019). More specifically, we multiply the accumulated gradient of alpha, $\nabla_{\alpha_l}$, with $1/\sqrt{|W_l|}$, where $|W_l|$ represents the cardinality of the weight

matrix, *i.e.* the number of weights in layer $l$. This scaling promotes a more controlled and slower update of the quantization range, while still allowing the network to learn the optimal $\alpha_l$ values.

To further promote a small $\alpha$, we add an L2 regularization term to the $\alpha$ update at each iteration. The strength of the regularization is controlled by $\lambda$, the regularization coefficient, and the added loss term is expressed by $\frac{\lambda}{2} \sum_l \alpha_l^2$. The trade-off between robustness and performance can be controlled by adjusting the value of $\lambda$. A large value of $\lambda$ results in an increasingly smaller $\alpha$ during training and thus a more robust network since the quantization range will be reduced. On the other hand, a smaller value of $\lambda$ will result in a larger $\alpha$ and a less robust network but with better noiseless performance.

### 3.3 Straight-through gradient estimation

After applying the clipping function (equation 2), a possible approach to update the weights during back-propagation is to set the gradient to 1 for the weight values that lie inside the clipping range and to 0 for the values outside the range which is known as piece-wise linear (PWL) gradient estimator (Baskin et al., 2021):

$$\frac{\partial \hat{w}_l}{\partial w_l} = \begin{cases} 1, & \text{if } w_l \in [-\alpha_l, \alpha_l] \\ 0, & \text{otherwise} . \end{cases} \tag{6}$$

Using this approach, the weights outside the range are not updated, and instead, the scaling factor is adjusted to bring the dequantized value closer to its optimal value. However, in our case, we want to decrease the scaling factor $\alpha$ to apply aggressive weight clipping and promote DNN robustness. Therefore, we propose to set the gradient to 1 for all weight values $W$, regardless of whether they lie inside or outside the quantization range. This is achieved using the straight-through gradient estimation (STE) technique, which allows the gradient to flow through the clipping operation without being affected.

### 3.4 Bit-flip noise (BFN) injections

To further improve DNN robustness at inference time, a popular approach is to simulate bit-flip errors during training (Henwood et al., 2020; Stutz et al., 2021). More specifically, let $f_{\{W_l\}_{l=1}^L}(X)$ be a network with weights $\{W_l\}_{l=1}^L$ and $(X, Y)$ be a training input and corresponding label from the training set. The desired objective to minimize may be defined as:

$$l = \mathbb{E}_\epsilon \left[ \mathcal{L} \left( f_{\{\tilde{W}_l\}_{l=1}^L}(X), Y \right) \right], \tag{7}$$

where $\mathcal{L}$ is the loss function and $\tilde{W}_l$ is a perturbed representation of the weights $W_l$ obtained as follows:

$$\tilde{W}_l = \frac{\alpha_l}{2^{b-1} - 1} \times \left( W_l^{\text{int}} \oplus \epsilon \right), \tag{8}$$

where $W_l^{\text{int}}$ is the quantized representation of $W_l$, $\oplus$ denotes the bitwise XOR operation, $\epsilon = \sum_{i=0}^b 2^i \epsilon_i$ with $\epsilon_i \sim \text{Bernoulli}(p_o)$, and $p_o$ is the bit error rate. In other words, the desired objective is the average loss of the network over all possible noise realizations.

However, this objective is intractable due to the exponentially large number of possibilities. Moreover, the XOR operation is not differentiable and, therefore, needs to be approximated. A common approach for approximating the objective in equation 7 is to use STE for the noise-injecting model, which passes the output gradients through the input gradients. Hence, by sampling $\epsilon$ from the noise distribution and using one noise realization per minibatch, assuming stochastic minibatch optimization, we can complete the forward pass by injecting the bit-flip noise to the weights to compute the perturbed weights $\tilde{W}_l$ and then compute the backward by setting $\frac{\partial \tilde{W}_l}{\partial W_l} = 1$. Using STE ultimately enables us to backpropagate gradients through the non-differentiable process of adding noise to the weights, making it possible to minimize the objective.

---

**Algorithm 1** Weight clipping-aware training (WCAT)

---

**Input:** bit error rate $p_o$, alpha weight decay $\lambda$, learning rate $\gamma$
 1: **for** l = 1 to $L$ **do**
 2:     initialize $\alpha_l$ using equation 10
 3: **end for**
 4: **for** t = 1 to $T$ **do**
 5:     sample batch (X,Y) from training data
 6:     **for** l = 1 to $L$ **do**
 7:         $\bar{W}_l = \text{Clip}(W_l, -\alpha_l, \alpha_l)$
 8:         $W_l^{\text{int}} = \text{Quantize}(\bar{W}_l, \alpha_l)$
 9:         $\epsilon = \sum_{i=0}^{b} 2^i \epsilon_i, \quad \epsilon_i \sim \text{Bernoulli}(p_o)$
10:         $\hat{W}_l^{\text{int}} = W_l^{\text{int}} \oplus \epsilon$
11:         $\tilde{W}_l = \text{De-quantize}(\hat{W}_l^{\text{int}}, \alpha_l)$
12:     **end for**
13:     $\Delta = \nabla \mathcal{L}\left(f_{\tilde{W}}(X), Y\right) + \lambda \sum_{l=1}^{L} \alpha_l$
14:     $W^{t+1} = W^t - \gamma \Delta$
15: **end for**

---

### 3.5 Quantization range initialization

We can search for a quantization range $\alpha^*$ that minimizes the quantization error, for a given weight distribution. This optimal range $\alpha^*$ can be determined by minimizing the following equation:

$$\alpha^* = \arg\min \alpha_w \left(W - DQ\left(W_{int}\right)\right)^2 . \tag{9}$$

Here, $DQ()$ represents the dequantization step specified in Equation 4. Additionally, we can find the optimal range considering the expected quantization error under a given bit error rate $p_o$. This can be achieved by minimizing the following equation:

$$\alpha^* = \arg\min \alpha_w \mathbb{E}_\epsilon \left[\left(W - DQ\left(W_{int} \oplus \epsilon\right)\right)^2\right] . \tag{10}$$

In the context of using pretrained models, we observed that initializing weights is crucial. By initializing the quantization range of each layer using equation 10 with a brute force search, we introduced a negligible overhead at the beginning of the training process. However, this initialization greatly improved both the robustness and performance of the model.

### 3.6 Summary of WCAT

We outline our proposed approach in Algorithm 1, which encapsulates the steps of WCAT. First, we initialize the quantization parameters $\alpha_l$ heuristically (*line 2*). During each training iteration's forward pass, we quantize the clipped weights (*lines 7 and 8*), inject bit-flip noise (*line 10*), and then dequantize the perturbed weights (*line 11*). Subsequently, we compute the gradient of the loss with respect to the perturbed weights and the regularization term (*line 13*). Finally, we update the model parameters utilizing straight-through gradient estimation (*line 14*). In essence, WCAT dynamically adjusts quantization ranges during training to strike a balance between achieving better performance and improving robustness.

## 4 Experimental results

In this section, we present the experimental results of the proposed WCAT method. We begin by outlining our experimental setup in Section 4.1. Then, we investigate the impact of regularization (Section 4.4) and gradient scaling (Section 4.5) on model robustness. We also explore the effects of initializing the quantization range for fine-tuning pre-trained models in Section 4.7. Next, we compare the different methods in terms of bit-error robustness against random (Section 4.2) and adversarial attacks (Section 4.3). Furthermore, we

demonstrate how our proposed method can significantly reduce energy consumption at inference time while maintaining a desired level of reliability in Section 4.8. Finally, we investigate the reasons behind the notable improvement of the proposed method when using pre-trained weights in Section 4.9.

## 4.1 Experimental settings

We conducted our experiments using three popular vision datasets: CIFAR10, CIFAR100 (Krizhevsky et al., 2009), and ImageNet (Krizhevsky et al., 2017). For our experiments, we employed ResNet-20, ResNet-56, and ResNet-18 architectures for CIFAR10, CIFAR100, and ImageNet, respectively. We also used data-efficient image Transformers (DeiT) (Touvron et al., 2021) to showcase the effectiveness of our method on vision transformer models. The evaluation metrics included measuring the robustness against random bit-flip and adversarial attacks. We compared our proposed method, WCAT, against baselines including linear quantization (Normal), per-layer weight clipping (PLClip), and LSQ. Detailed information on the datasets, training setups, noise injection, evaluation metrics, and baselines can be found in Appendix B.

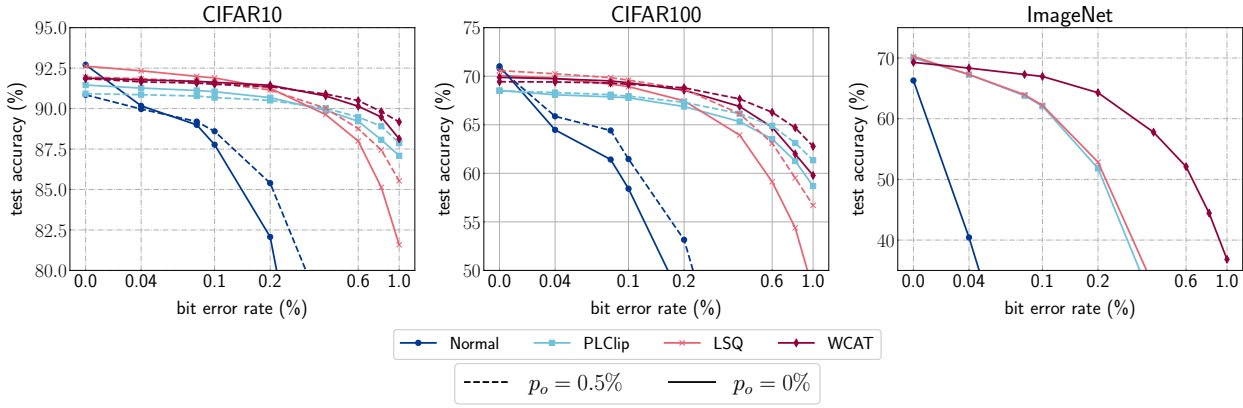

Figure 1: Robustness of different quantization methods on ResNets with 4-bit weight quantization.

## 4.2 Random Attack

We compare the bit-error robustness of different methods for 4-bit weight quantization on ResNet-20 trained on CIFAR10, ResNet-56 trained on CIFAR100, and ResNet-18 on ImageNet in Figure 1. Overall, WCAT is the superior method, as it maintains high performance across all tested noisy regimes on all models and datasets. Across all datasets, LSQ achieved the best accuracy on noiseless models. In CIFAR10, LSQ and WCAT perform similarly under low noise conditions, but WCAT shows significantly better performance in high-noise regimes. However, we observe a significant improvement in ImageNet results. Even with a small amount of noise (e.g., $10^{-4}$), the accuracy of normal quantization drops dramatically. Although LSQ and PLCLip show better robustness than normal quantization, WCAT outperforms all methods in highly noisy regimes.

In all methods, bit-flip noise (BFN) injection training tends to improve robustness, especially under highly noisy regimes. Although BFN significantly sacrifices test accuracy for normal quantization under noiseless conditions, this performance loss is negligible in PLCLip and WCAT. We also present results for 8-bit quantization in Figure 2. We see that LSQ exhibits the best accuracy on noiseless weights for all models. However, when faced with even a minor bit-flip noise, LSQ's accuracy drops significantly. In contrast, WCAT demonstrates remarkable robustness across all datasets. Additionally, we observe that ImageNet on 8-bit is more sensitive to bit-flip noise than in 4-bit.

We also investigated the impact of random bit-flips in DeiT with 8-bit weight quantization on ImageNet. The results are shown in Figure 3. In this experiment, we set the PLClip parameter to 0.4 for DeiT-tiny and 0.6 for DeiT-small and DeiT-base models. Once again, WCAT outperforms all the other methods in all variations of DeiT, with the performance difference becoming more significant as the model size increases.

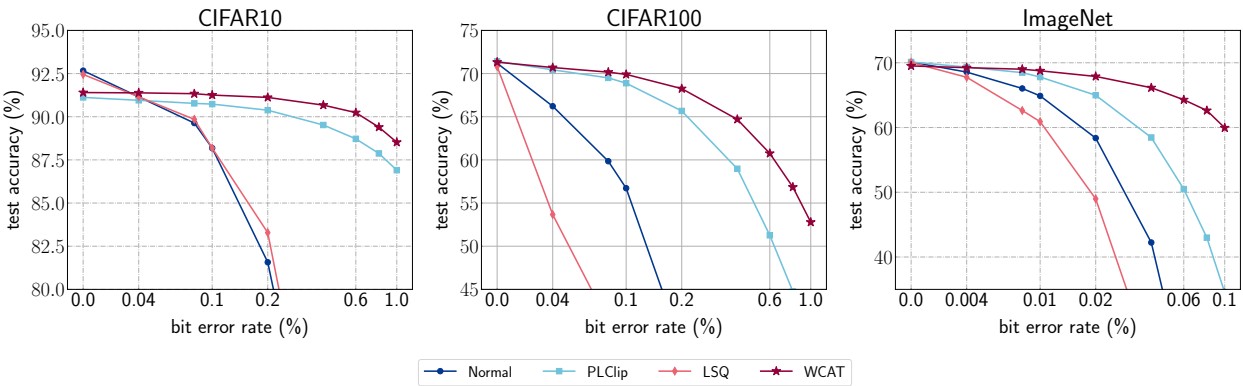

Figure 2: Robustness of different quantization methods on ResNets with 8-bit weight quantization.

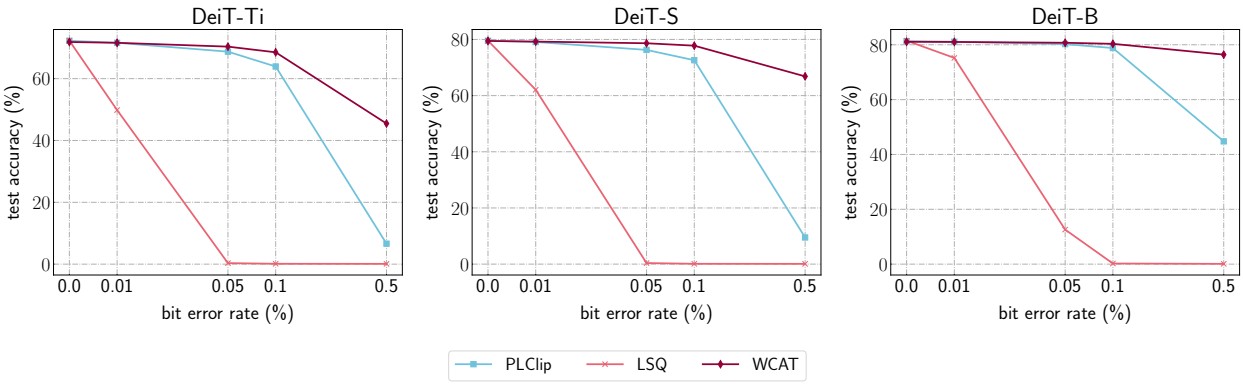

Figure 3: DeiT robustness under various 8-bit weight quantization methods on ImageNet.

### 4.3 Adversarial Attack

We now focus on an additional scenario: robustness to adversarial attacks. The robustness of different methods against adversarial bit-flip attacks for all models and for both 4-bit and 8-bit quantization is presented in Figure 4. For each dataset, we show the average number of bit-flips required to destroy network performance and reduce accuracy to random guesses over 10 different seeds for the attack method as we described in Appendix B.

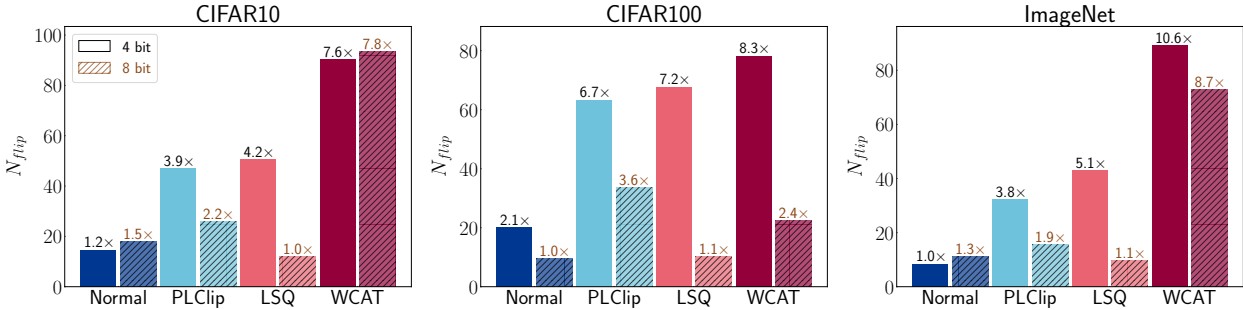

Figure 4: Impact of adversarial bit-flip attacks on network performance using ResNets with 4 and 8-bit weight quantization on CIFAR10, CIFAR100, and ImageNet. $N_{flip}$ denotes the average number of bits required to degrade top-1 accuracy to 0.1%, 1%, and 10% for ImageNet, CIFAR100, and CIFAR10, respectively.

Notably, we observe that WCAT not only enhances robustness against random noise but also outperforms all methods in all models against adversarial attacks under 4-bit quantization. This suggests that our proposed method benefits extend beyond improving network robustness in the presence of random noise. Moreover, WCAT's superiority is apparent in ImageNet results, where it requires $10.6\times$ more bit-flips to cause network performance degradation when compared to other methods.

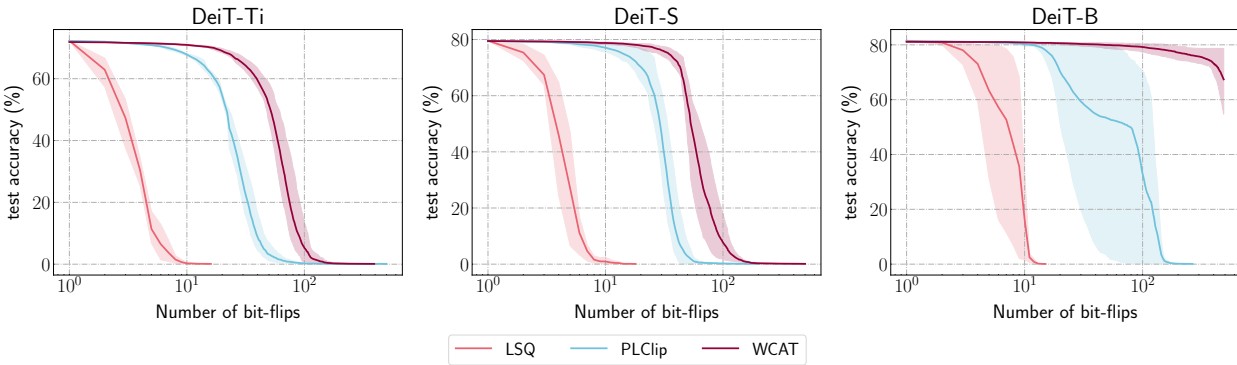

Figure 5: Robustness comparison of 8-bit weight quantization methods on DeiT against adversarial attacks on ImageNet.

Additionally, we explore the effect of adversarial attacks on DeiT in Figure 5. Surprisingly, the vision transformer exhibits greater resilience to adversarial bit-flip attacks on the weights for both WCAT and PLClip. Notably, WCAT consistently outperforms other methods across all DeiT variations and demonstrates significant robustness, particularly in the DeiT-base model. Even after 500 bit-flip attacks, the average accuracy remains above 70%. This demonstrates the effectiveness of WCAT in protecting networks from adversarial attacks in addition to random noise.

## 4.4 Regularization

We start by investigating the impact of L2 regularization of the quantization ranges on the robustness and performance of different models in the presence of noise. Particularly, we vary the L2 regularization coefficient and evaluate the resulting models under different noise realizations. The results, encompassing average test accuracy alongside standard deviation, are presented in Table 1, showing that increasing the L2 regularization tends to improve robustness in noisy environments, at the expense of decreased performance in the noiseless setting.

Specifically, we find that increasing the regularization strength to force the model to have a lower quantization range can improve robustness, as evidenced by the results for $\lambda = 0.5$ for the ImageNet dataset. However, this comes at a significant cost in terms of performance in the absence of noise. In these scenarios, it may be more beneficial to train the model with bit-flip noise to improve its robustness against noise.

## 4.5 Gradient Scaling

We now investigate the impact of scaling gradient on the performance and robustness of the models in Table 1. Our findings indicate that with low regularization, the scaling gradient not only improves the performance of the noiseless model but also enhances the robustness of the noisy regime for both models. Interestingly, the addition of an L2 regularizer further improves the performance of the models with a gradient scale of 1. As discussed in Section 3.2, updating the quantization range involves accumulating all the gradients of weights at each layer with respect to $\alpha$, which can lead to high gradients and instability without gradient scaling. The addition of a regularizer helps to restrict the rapid change when we don't use gradient scaling. Ultimately, we observe that the models trained with scaled gradients achieve the best performance for different levels of noise. These results suggest that scaling gradient besides increasing regularization can significantly improve the robustness of the model against bit errors, especially in noisy environments, while enhancing the overall performance of the model.

Table 1: Robustness of ResNet-20 on CIFAR10 and ResNet-18 on ImageNet with 4-bit quantization for different gradient scales and L2 regularization coefficients. The average test accuracy and standard deviation of models under different noise realizations are reported. The 32-bit floating-point baseline's test accuracy is 92.6% and 69.75% for ResNet-20 and ResNet-18, respectively.

| Model | Grad scale | Reg. coeff. $\lambda$ | Test accuracy (%) | | |
|---|---|---|---|---|---|
| | | | $p_o = 0\%$ | $p_o = 0.1\%$ | $p_o = 1\%$ |
| ResNet-20 on CIFAR10 | 1 | 0.0005 | 90.77 | $83.06_{\pm 3.6}$ | $17.8_{\pm 4.9}$ |
| | | 0.001 | 91.18 | $86.49_{\pm 2.3}$ | $23.02_{\pm 6.4}$ |
| | | 0.01 | 92.36 | $91.37_{\pm .3}$ | $75.76_{\pm 5.1}$ |
| | $1/\sqrt{N}$ | 0.0005 | **92.48** | $91.55_{\pm .35}$ | $77.72_{\pm 5.4}$ |
| | | 0.001 | 92.41 | $\mathbf{91.93_{\pm .2}}$ | $83.42_{\pm 3.4}$ |
| | | 0.01 | 91.88 | $91.69_{\pm .1}$ | $\mathbf{88.62_{\pm 1.2}}$ |
| ResNet-18 on ImageNet | 1 | 0.001 | 68.59 | $31.46_{\pm 5.9}$ | $0.1_{\pm .01}$ |
| | | 0.01 | 69.87 | $51.73_{\pm 4.4}$ | $0.2_{\pm .07}$ |
| | | 0.5 | 69.91 | $66.36_{\pm 1.1}$ | $22.84_{\pm 5.2}$ |
| | $1/\sqrt{N}$ | 0.001 | **70.31** | $65.91_{\pm 1.14}$ | $16.04_{\pm 5.61}$ |
| | | 0.01 | 69.79 | $\mathbf{67.76_{\pm .75}}$ | $42.85_{\pm 4.88}$ |
| | | 0.5 | 66.56 | $65.11_{\pm .5}$ | $\mathbf{47.26_{\pm 3.4}}$ |

## 4.6 Gradient Estimation

In Section 3.3, we explained how we used STE to update the weights that are both inside and outside the clipping range during training. Sakr et al. (2022) argued that STE may encounter gradient explosion, while the use of PWL can lead to partial convergence halt. To address these concerns, they present a novel magnitude-aware derivative (MAD) gradient estimator.

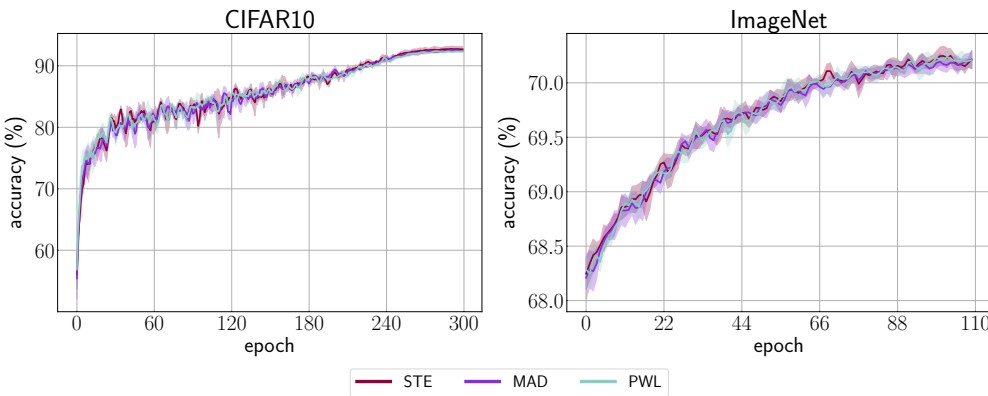

Figure 6: Comparative evaluation of gradient estimation techniques using ResNet-20 on CIFAR10 and ResNet-18 on ImageNet datasets with 4-bit quantization. The plot showcases validation accuracy, with the average accuracy represented by a solid line, and the shaded region signifying standard deviation across 5 seeds.

We extensively examined these techniques, as depicted in Figure 6, employing ResNet-20 on CIFAR10 and ResNet-18 on ImageNet. The assessment involved 4-bit weight quantization with a constant regularization value of $\lambda = 0.001$. This process was repeated across 5 distinct seeds, with the average accuracy represented by a solid line and the standard deviation displayed as a shaded area adjacent to the line. Our findings reveal that these approaches exhibit comparable performance in terms of validation accuracy during training/finetuning. Given its straightforward implementation, we opt for STE to estimate the gradient of the weights.

Table 2: Comparison of robustness for ResNet-18 on ImageNet using different quantization range initialization for 8-bit weight quantization.

| Model | Initialization method | Test accuracy (%) | |
|---|---|---|---|
| | | $p_o = 0\%$ | $p_o = 0.1\%$ |
| ResNet-18 | Max_Abs | 70.09 | $8.6_{\pm 3.1}$ |
| | Optimal MSE | **70.10** | $22.6_{\pm 6.26}$ |
| | Optimal Expected MSE | 69.51 | $\mathbf{59.96_{\pm 2.53}}$ |

### 4.7 Quantization Range

In Section 3.5, we outlined our approach to initializing the quantization range. Our method involves performing a brute force search to find the optimal quantization range that minimizes the expected MSE between the real weights and the dequantized weights, considering the presence of bit-flip noise. In contrast, a simpler approach, referred to as the Max_Abs method, initializes the range based solely on the maximum absolute value of the weights. To assess the impact of these initialization methods, we conducted experiments on ResNet-18 fine-tuned on ImageNet and present the results in Table 2.

We observe that both the Max_Abs and optimal MSE methods yield better noiseless performance for 8-bit quantization. However, this comes at a cost to the robustness of the network after finetuning, as the larger initial range of the weights can lead to increased sensitivity to perturbations. Notably, initializing the quantization range using our proposed method for a specific bit error rate can significantly enhance the robustness of the fine-tuned model.

### 4.8 Energy-reliability trade-off

Lowering the memory supply voltage reduces energy consumption, especially in SRAMs, but it can introduce bit-flip errors and therefore reduces the reliability of DNN accelerators. Hence, enhancing DNN robustness becomes crucial for ensuring resilience in low-power applications. To this end, we model the trade-off between energy consumption and reliability in SRAMs by applying quadratic regression on the data from (Di Mauro et al., 2020, Fig. 11), which is based on measurements on fabricated chips. Detailed information concerning the energy model is expounded in Appendix C.

For each DNN model, we use Accelergy Wu et al. (2019) to estimate the energy consumption on the Eyeriss accelerator Chen et al. (2016), thus including both memory-access and operation energy. We modified the Eyeriss architecture to have only on-chip memory, implemented with SRAM, for all data types. We assume only the DNN weights to be stored unreliably, with the activations and partial sums being stored in a reliable SRAM. We also assume that batch normalization layers are fused with the convolutional layers. The energy consumption at nominal voltage (0.8 V) for one image (batch size of 1) is $118.37\,\mu\mathrm{J}$, $285.96\,\mu\mathrm{J}$, $390.54\,\mu\mathrm{J}$, and $495.12\,\mu\mathrm{J}$ for ResNet-20, ResNet-32, ResNet-44, and ResNet-56, respectively. Importantly, we note that these nominal-voltage runs are not Pareto optimal and are consequently excluded from the figure for clarity.

We present the accuracy versus energy consumption of the aforementioned ResNet models on CIFAR-10 using the different methods and levels of noise in Figure 7. All methods are repeated with and without BFN. We show the points in the Pareto frontier in specified color, which represents the best trade-off between energy consumption and accuracy. We observe that WCAT almost always achieves the lowest energy consumption for a given accuracy level. This highlights the ability of WCAT to significantly reduce energy consumption during inference without compromising much reliability compared to existing methods.

### 4.9 Effect of pre-trained weights

As previously discussed, WCAT exhibits a significant improvement in robustness on ImageNet for both random bit-flip and adversarial attacks across all models. In order to investigate this observation further, we conducted an experiment to examine the impact of finetuning pre-trained models. We initiated the training

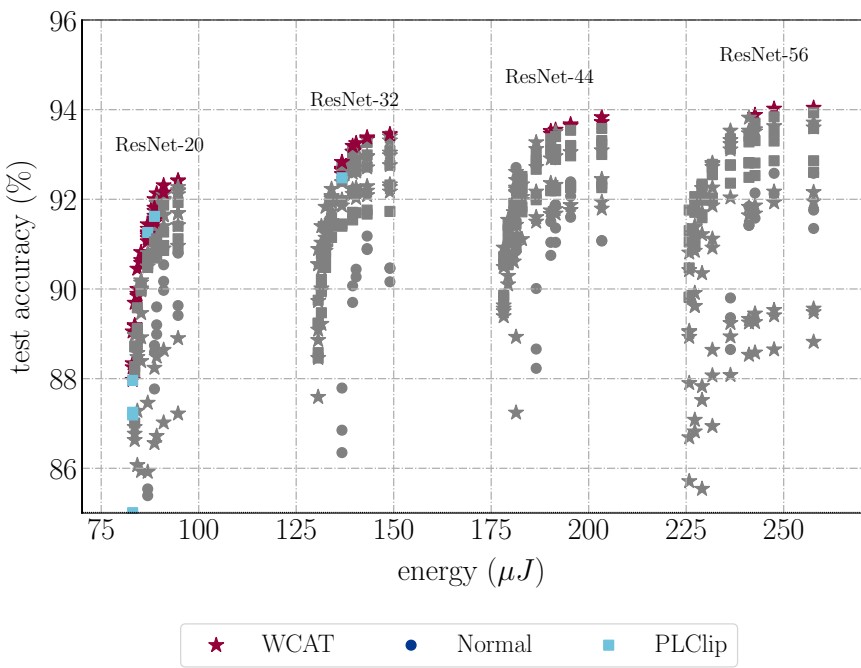

Figure 7: Classification accuracy in terms of energy per inference run on CIFAR10.

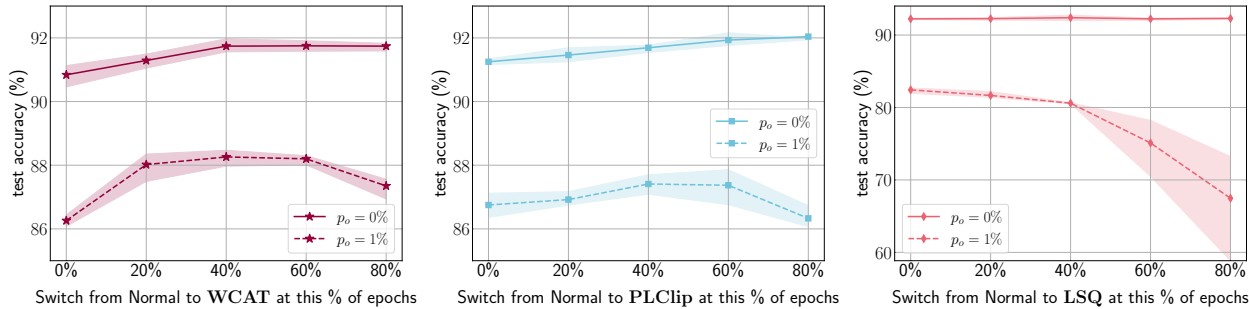

Figure 8: Impact of finetuning on accuracy for noiseless and 1% bit error rate models of ResNet-20 on CIFAR10.

process with normal training and, after a certain number of epochs, switched to the WCAT quantization method. We repeated this process for 3 different seeds. Figure 8 illustrates the average accuracy as a solid line and the minimum and maximum performance as a shaded area for both noiseless models and models subjected to BFN training with a $p_o = 1\%$ bit-flip probability, specifically for ResNet-20 on CIFAR10.

We see that for WCAT when changing the quantization method in later epochs and attempting to finetune the model, the accuracy of both noiseless and noisy models increases. On the other hand, for LSQ, starting from the beginning of the training process is crucial. While PLClip shows similar behavior to WCAT, changing to PLClip later can slightly increase both performance and robustness. However, when varying the regularization coefficient ($\lambda$) in WCAT (Figure 9), this change becomes more significant for higher values of $\lambda$. This aligns with our observation on ImageNet results, where using higher $\lambda$ with pre-trained weights is viable, while training from scratch on CIFAR10 and CIFAR100 with high lambda may not converge. This observation provides insight into why WCAT performs significantly better on pre-trained models compared to other methods.

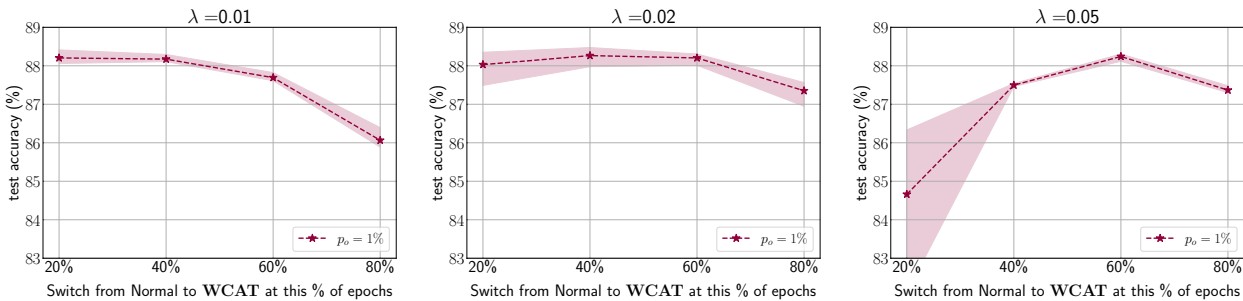

Figure 9: Impact of finetuning on accuracy for noiseless and 1% bit error rate models of ResNet-20 on CIFAR10.

## 5 Conclusion

In this paper, we presented a robust quantization scheme based on a key observation that emphasizes the crucial role of the quantization range in enhancing the robustness of DNNs. Our method incorporates a learning mechanism to optimize the quantization range of different layers during training, achieving a delicate balance between robustness and noiseless performance. Our experimental results across different architectures and datasets showcase that WCAT outperforms existing robust quantization techniques, exhibiting higher accuracy levels and better defense against both random and adversarial attacks. In the end, we show that WCAT consistently provides a practical solution to improving the resilience of DNNs against bit-flip errors.

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

## A  Optimal Quantization Range

To motivate the connection between quantization range and robustness, we provide an empirical analysis of the effect of random bit error noise on the optimal quantization range of a simple multi-layer perceptron (MLP) with a single hidden layer and 100 neurons trained on the MNIST dataset. We use cross-entropy as the objective function and use the same quantization range for the first and second fully connected layers. We evaluate our toy model on the test set for different levels of noise, which we introduce in Section 3.4. To estimate the exact loss of each trial, we use Monte-Carlo estimation, which involves sampling random instances of the noise and computing the average loss. Results are presented in Figure 10a.

Our results indicate that increasing the bit error rate leads to a smaller optimal quantization range. This finding highlights the importance of our proposed approach, as it ensures that the network maintains its performance under noiseless conditions while achieving a more robust network against bit-flip errors during inference.

Also, following a similar method in Choi et al. (2019), in Figure 10b we demonstrate the $\alpha^*$ on the (y-axis) versus bit error rate (x-axis) for some commonly known weight distributions such as $\mathcal{N}(0,1)$, $U(-1,1)$,

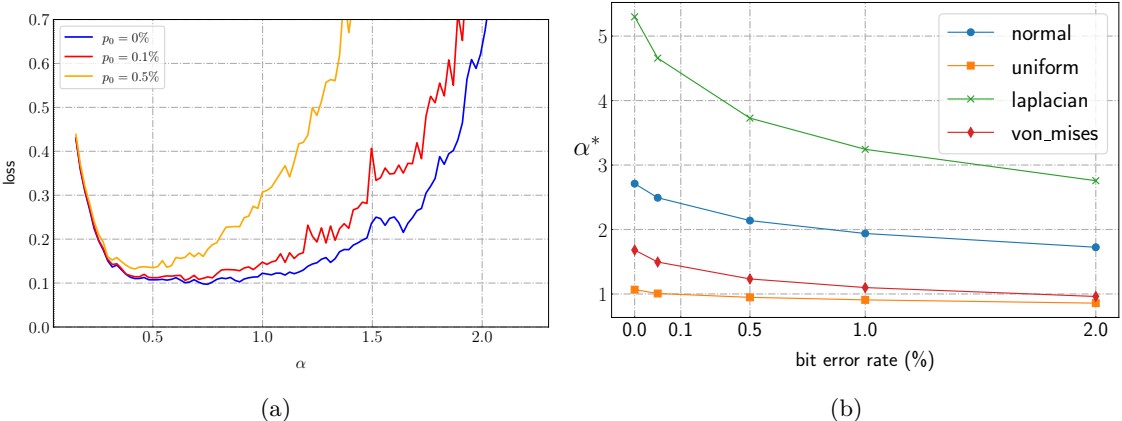

(a)                                                    (b)

Figure 10: Optimal quantization range under bit error rate noise (a) Optimal quantization range for different levels of noise on a simple MLP with one hidden layer trained on the MNIST dataset. (b) Optimal quantization range from different well-known distributions for varying levels of noise.

Laplacian$(0, 1)$, and Von mises$(0, 4)$, where the $\alpha^*$ is derived heuristically based on Equation 10. Each point represents the optimal expected range for a different level of noise. As we can see, as the level of noise increases, the optimal quantization range becomes smaller, emphasizing the importance of our proposed approach to reduce the quantization range.

## B    Experimental Setups

We provide additional details about our experimental results in this appendix.

### B.1    Datasets

Both CIFAR10 and CIFAR100 comprise 60,000 RGB images with dimensions of $32 \times 32 \times 3$. As per the standard approach, 50,000 examples were utilized for training and the remaining 10,000 for testing, with images evenly drawn from 10 and 100 classes, respectively. On the other hand, ImageNet is a more extensive dataset, consisting of 1.2 million training images across 1,000 classes, with input dimensions of $224 \times 224 \times 3$. Our work employs the same data augmentation techniques as those in He et al. (2016).

### B.2    Training Setups

Each model on CIFAR10 and CIFAR100 was trained using stochastic gradient descent (SGD) Robbins & Monro (1951) with a Nesterov momentum of 0.9, a weight decay of $5 \times 10^{-4}$, and a batch size of 128 for 300 epochs for 4 and 8 bit weight quantization. We used a cosine annealing learning rate scheduler, with an initial rate of 0.1.

To fine-tune our 8-bit network on ImageNet, we began with pre-trained weights and conducted a single epoch of additional training, using SGD with momentum and a learning rate of $10^{-4}$. As for our 4-bit networks, we initialized the pre-trained weights and fine-tuned them for 110 epochs using SGD with momentum, with a batch size of 256. We implemented exponential learning rate decay, starting from an initial rate of 0.0015 and decaying it to a final value of $10^{-6}$.

### B.3    Noise Injection Training

We conducted experiments applying various levels of bit-flip noise during training, as detailed in Section 3.4, in addition to the previous methods. Our results showed that a low bit error rate during training did not improve the model's robustness during inference. Conversely, a high amount of noise completely destroyed

the network's performance during training and prevented it from converging. After testing different bit-flip probabilities $p_o$ ranging from $10^{-4}$ to $10^{-2}$, we found that a value of $p_o = 5 \times 10^{-3}$ increased the model's robustness, with only a slight degradation in performance in the absence of noise. Furthermore, in the case of ImageNet, the network proved highly sensitive to minor perturbations, making it challenging to converge, even with a small amount of noise during training.

### B.4 Evaluation Metrics

In order to assess the robustness of DNNs, we performed 50 iterations of bit-flip noise for each test example, and the classification result was obtained by averaging the outcomes. For evaluating the resilience of the DNNs against adversarial attacks, we modified the code from Rakin et al. (2019) and used the number of bit-flips ($N_{flip}$) as the metric to measure BFA resistance, which corresponds to a reduction of top-1 accuracy to below 11%, 1.1%, and 0.11% for CIFAR10, CIFAR100, and ImageNet respectively. Since BFA requires a set of data to perform the attack, we randomly sample 64 images from the test set as the default configuration. Subsequently, we calculate the test accuracy using the remaining data and report the average value of $N_{flip}$ across 10 different seed values.

Note that, the proposed BFA is performed through randomly draw a sample of input images from the test/validation set, where the default sample size is 128 and 256 for CIFAR-10 and ImageNet respectively. Then, only the sample input is used to perform BFA, where the rest data and ground-truth labels are isolated from the attacker. Moreover, each experimental configuration is run with 5 trials to alleviate error caused by the randomness of sampling input.

### B.5 Baselines

We conducted a comparison between our proposed method, WCAT, and three baselines, including linear quantization (Normal), per-layer aggressive weight clipping (PLClip) - which is one of the state-of-the-art methods for dealing with bit-flip errors - and LSQ, a state-of-the-art quantization method with learnable step size, using both 4-bit and 8-bit weight quantization. Notably, we made modifications to the PLClip method by applying parameter clipping exclusively to linear and convolutional layers. The reasoning behind this modification and our observations are detailed in Appendix D.

To determine the optimal standard clipping value for PLClip, we conducted a hyperparameter search within the range of 0.01 to 0.8 and we found the best standard clipping value for PLClip to be 0.1 for ResNet-20 on CIFAR10, 0.3 for ResNet-56 on CIFAR100, and 0.7 for ResNet-18 on ImageNet. It's worth noting that to generate a different clipping range per layer, we multiplied the clipping value by the ratio between the maximum absolute weight of the given layer and the maximum absolute weight across the entire model. However, it's crucial to mention that PLClip is highly sensitive to the clipping hyperparameter selection, and finding the best value is a time-consuming process. Additionally, computing the clipping range for each layer using PLClip after each iteration significantly increases the training time.

## C Energy model

As discussed earlier, Di Mauro et al. (2020) offers a detailed exploration of fabricated SRAMs and their bit error rates under ultra-low energy conditions using Global Foundries 22nm FDX technology. They found that as the supply voltage decreases, the bit error rate (BER) rises exponentially. By performing a quadratic regression on the experimental data, we obtain the following BER model:

$$\epsilon = \min \left( \exp \left[ a + bV + cV^2 \right], 0.5 \right), \tag{11}$$

where $V$ represents the supply voltage and specific constants are defined as follows: $a = 22.12$, $b = -68.14$, and $c = 0$. Since dynamic power is proportional to the square of supply voltage, we obtain the following energy model:

$$E = \frac{V^2}{V_{\text{nom}}^2} \cdot E_{\text{nom}}, \tag{12}$$

where $V_{\mathrm{nom}} = 0.8$V refers to the nominal supply voltage, and $E_{\mathrm{nom}}$ represents the energy consumption of the memory at the nominal voltage, as reported by CACTI (Li et al., 2011).

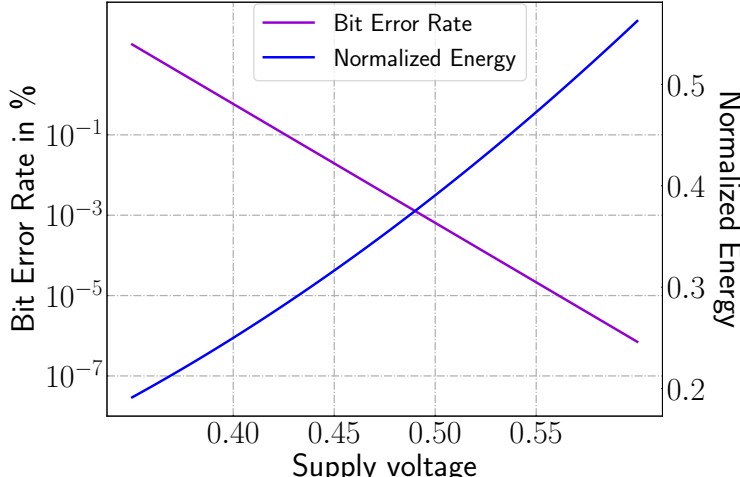

Figure 11: Bit error rate measurements from Di Mauro et al. (2020) show the impact of supply voltage scaling on SRAM memories.

Figure 11 visually explains what our energy model means. It shows that lowering the memory supply voltage saves energy, but at the same time, increases the likelihood of bit flips. This insight serves to emphasize the critical balance between energy efficiency and reliability.

# D    PLClip and batch normalization layers

In Stutz et al. (2022), the authors originally suggest applying weight clipping to all layers, including batch normalization layers. Here, we show that restricting the clipping operation to only convolutional and linear layers can enhance model robustness. Figure 12 illustrates the comparison between the main PLClip method and the modified version against random noise. We specifically evaluate the performance of ResNet-20 on CIFAR10 and ResNet-18 on ImageNet, focusing on 4-bit quantization during training with and without bit-flip noise (BFN). The results demonstrate that by not clipping the weights in the batch normalization layers, we obtain a better accuracy and bit-error rate trade-off.

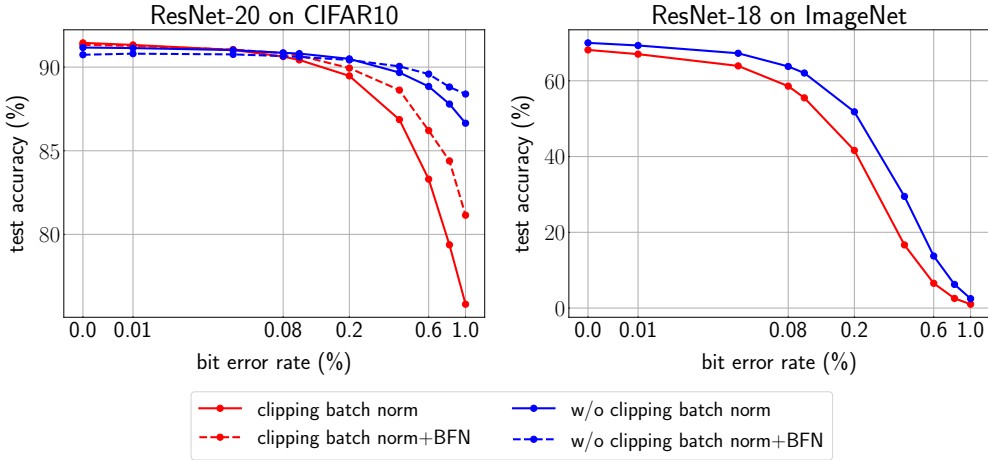

Figure 12: Comparison of robustness between applying and not applying PLClip on batch normalization layers for 4-bit quantization.

# E Adversarial training

In this section, we introduce adversarial training as a baseline for comparison with our proposed method. Adversarial training has been explored in prior works (Stutz et al., 2022; He et al., 2020), and we adopt the same objective proposed in Adversarial Bit Error Training (AdvBET) as described by (Stutz et al., 2022). The objective function is defined as

$$\min_{W} \left( \mathcal{L} \left( f_{\hat{W}}(X), Y \right) + \mathcal{L} \left( f_{\tilde{W}_{BFA}}(X), Y \right) \right), \tag{13}$$

where $\hat{W}$ represents the dequantized weights, and $\tilde{W}_{BFA}$ denotes the dequantized version of the weights perturbed under the BFA, following the objective described in equation 1.

We follow a setup similar to that outlined by Stutz et al. (2022), starting adversarial training when the loss falls below 1.75 on CIFAR-10. Additionally, we apply gradient clipping to the faulty forward pass, constraining the gradients to the range $[-0.05, 0.05]$. For rest of the training setup, we use the same training recipe as discussed in Appendix B.2.

It is important to note that adversarial training significantly extends the training time due to the repeated execution of the BFA at each training iteration. For example, when performing 5-bit flips using BFA at each iteration, training ResNet-20 on CIFAR-10 requires approximately $16.5\times$ more time compared to standard training without adversarial components. For this reason, we only introduce BFAs after completing 50% of the total epochs when performing 10-bit flip attacks to amortize the training time.

In Table 3, we present the clean accuracy and $N_{\text{flip}}$, which represents the average number of bit flips required to induce the network to make random guesses. These results are obtained using various quantization methods and training ResNet-20 on the CIFAR-10 dataset with 4-bit quantization. In the context of adversarial training, we employ bit flip attacks with $k = 2, 5$, and $10$ at each training iteration, where $k$ is the number of adversarial bit flips selected by BFA.

Table 3: Robustness Evaluation of ResNet-20 on CIFAR-10 with 4-Bit Quantization Using Various Quantization Methods and Training Strategies

| | | Normal | | PLClip | | WCAT | |
|---|---|---|---|---|---|---|---|
| | | Test accuracy (%) | $N_{flip}$ | Test accuracy (%) | $N_{flip}$ | Test accuracy (%) | $N_{flip}$ |
| Vanilla training | | 92.62 | $11.3_{\pm 5.9}$ | 91.45 | $46.9_{\pm 18.9}$ | 91.88 | $78.8_{\pm 15.5}$ |
| BFN | $p_o = 0.5\%$ | 90.57 | $14.5_{\pm 3.6}$ | 90.98 | $30.1_{\pm 13.2}$ | 91.85 | $\mathbf{90.2_{\pm 31.5}}$ |
| Adversarial training | $k = 2$ | 92.34 | $17.3_{\pm 6.7}$ | 90.98 | $78.4_{\pm 16.3}$ | 91.00 | $83.1_{\pm 37.4}$ |
| | $k = 5$ | 92.43 | $17.9_{\pm 2.8}$ | 90.54 | $80.4_{\pm 24.4}$ | 91.02 | $65.4_{\pm 28.0}$ |
| | $k = 10$ | 91.90 | $33.8_{\pm 7.8}$ | 91.10 | $67.4_{\pm 27.2}$ | 91.84 | $61.9_{\pm 28.2}$ |

We first observe that adversarial training enhances the model's robustness against BFA using normal quantization. However, when compared to other robust quantization methods, it remains significantly more susceptible to adversarial attacks while also compromising clean performance. These findings suggest that adversarial training alone may not be considered a highly effective defense strategy, which aligns with the conclusions outlined by (He et al., 2020).

With PLClip, we observe a notable improvement in model robustness when employing adversarial training with 5 BFAs at each training iteration, achieving a two-fold increase in robustness. It's worth noting that while Stutz et al. (2022) also demonstrates the potential for adversarial training on top of per-layer clipping to enhance adversarial robustness, their specific method for injecting adversarial noise during training to perturb weights (AdvBET) differs from our baseline.

In AdvBET, rather than employing BFAs, the authors use their customized implementation of adversarial bit errors. Their approach initializes perturbed weights with the introduction of up to $\epsilon$ bits flipped, then

carries out gradient ascent steps in floating point to maximize the cross-entropy loss, followed by quantizing the weights to integer values and projecting the perturbed weights onto Hamming constraints. This iterative process is repeated for a specified number of iterations. The primary distinction between their method and BFA lies in the approach to progressively flipping bits, incorporating in-layer search and cross-layer search in an iterative manner. Due to the reported sensitivity of their training method to various hyperparameters, such as step size, gradient normalization, and momentum, coupled with the unavailability of their method's implementation to the public, we opted to utilize BFA as the basis for our baseline experiments.

Lastly, we observe that adversarial training yields only marginal improvements for WCAT, with the highest level of robustness still achieved by WCAT in combination with random flip noise injection during training.

## F  Targeted attack

In addition to assessing our proposed method against standard adversarial attacks, we expand our evaluation to include targeted adversarial attacks (Rakin et al., 2021a). These attacks are designed to manipulate a model's behavior in specific, malicious ways while preserving its expected performance for other inputs.

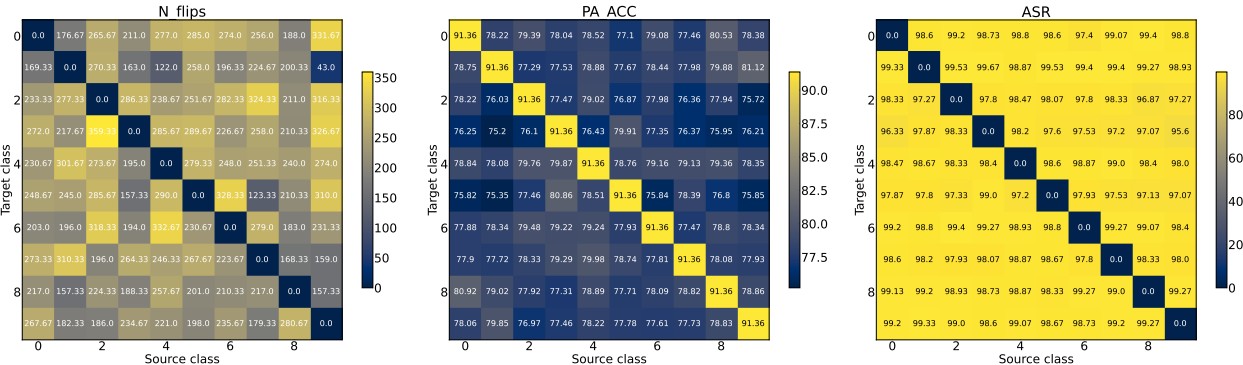

Figure 13: Performance evaluation of stealthy targeted adversarial attacks on ResNet-20 with 4-bit quantization: post-attack test accuracy, attack success rate, and average bit-flip counts across three attack rounds.

A notable benchmark for targeted adversarial attacks on weight-quantized DNNs is the targeted bit-flip attack (T-BFA) (Rakin et al., 2021a). Particularly, we evaluate the different methods using stealthy T-BFA, which is a variant of T-BFA designed with the goal of misclassifying all samples from a source class as if they belong to a target class. Importantly, this manipulation is executed while ensuring that test examples outside the attacked source class remain unaffected. As our evaluation metrics, we employ the attack success rate (ASR), post-attack test accuracy (PA-ACC), and the average number of bit flips used for each attack ($N_{flip}$), following (Rakin et al., 2021a).

Results using ResNet-20 models trained on CIFAR-10 with 4-bit quantization for WCAT are presented in Figure 13. Within this figure, we present three key metrics: $N_{flip}$, PA-ACC, and ASR, organized from left to right. These metrics represent the average values derived from three independent runs for each combination of source and target class within CIFAR-10. In summary, we observe that for certain combinations of source and target classes, the adversary requires more than 350 bits to achieve its objectives.

In Table 4, we present the evaluation of T-BFA on ResNet-20 with 4-bit quantization alongside the BFA. To assess the robustness of our proposed method, we compare it with another defense approach known as output code matching (OCM) (Özdenizci & Legenstein, 2022). OCM replaces traditional one-hot encoding with partially overlapping bit strings for multi-class classification, reducing attack stealthiness. In our evaluation, we utilize OCM with a code length of 64, a configuration that has demonstrated superior robustness for ResNet-20.

Our analysis reveals interesting insights into the performance of the OCM and our proposed method, WCAT, under targeted adversarial attack. We see that, while OCM demonstrates impressive resilience against

Table 4: Robustness Evaluation of ResNet-20 on CIFAR10: TBFA and BFA with 4-Bit Quantization.

| | Test Accuracy (%) | Stealthy T-BFA (Rakin et al., 2021a) | | | BFA |
| | | ASR (%)$\downarrow$ | $N_{flip}\uparrow$ | PA-ACC (%)$\downarrow$ | $N_{flip}\uparrow$ |
|---|---|---|---|---|---|
| Normal | **92.62** | 98.92 | 26.71 | 78.80 | 11.3 |
| WCAT | 91.88 | **98.46** | 237.38 | 78.09 | **90.2** |
| OCM-64 (Özdenizci & Legenstein, 2022) | 89.29 | 99.65 | **278.48** | **45.27** | 10.6 |

stealthy targeted attacks (as expected), it proves to be highly sensitive to BFA sometimes even underperforming standard quantization. This latter outcome is unsurprising since OCM primarily focuses on the final dense layer.

In a more in-depth analysis, we illustrate the distribution of layers affected by BFA and stealthy T-BFA in Figure 14 and 15, respectively. Notably, in targeted attacks, the bit-flip noise primarily impacts the last two layers, while in BFA, the first convolutional layer and intermediate layers are predominantly targeted.

In contrast, WCAT demonstrates exceptional robustness against stealthy targeted attacks, requiring only a marginal decrease in the $N_{flips}$ compared to OCM for inducing malfunction in the model. Impressively, WCAT showcases remarkable resilience against both BFA and stealthy T-BFA while managing to preserve the model's clean accuracy.

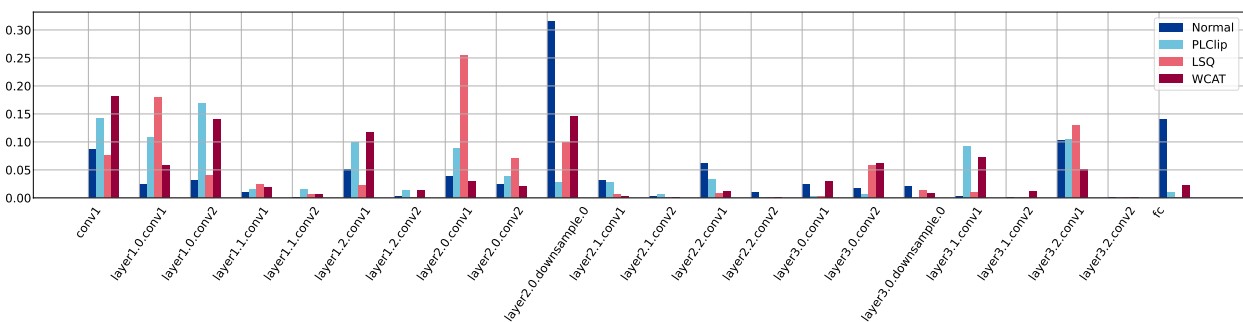

Figure 14: Distributions of attacked layers under BFA.

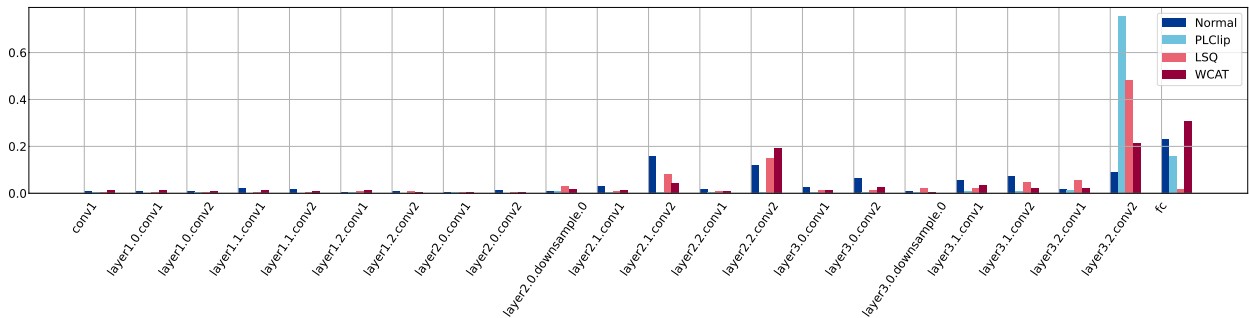

Figure 15: Distributions of attacked layers under Stealthy T-BFA.

