# OpenReview forum: "Training DNNs Resilient to Adversarial and Random Bit-Flips by Learning Quantization Ranges"
_TMLR — Accepted by TMLR_

### Review · Reviewer_eJU5 · 2023-09-05

**Summary Of Contributions:**

This paper introduces a symmetric signed quantization scheme, which achieves better robustness against bit-flip noise. Specifically, the weights are clipped and applied linear weight quantization. The clamping range is optimized and be regularized to a small range based on the empirical analysis. The authors conduct evaluation on various models and datasets to demonstrate the effectiveness of their algorithm.

**Audience:**

Yes

**Broader Impact Concerns:**

None.

**Claims And Evidence:**

Yes

**Requested Changes:**

1. Please justify the novelty.
2. Please clarify the motivation and provide more empirical evidence and analysis.
3. Please provide more evaluation with adversarial training against adversarial attacks.
4. Although this paper focuses on bit-flip noise, it could be interesting to see whether the proposed algorithm performs better robustness under traditional adversarial attacks on the image input, such as FGSM, PGD, etc.

**Strengths And Weaknesses:**

Strengths:
1. The proposed algorithm is easy to follow.
2. The authors conduct extensive evaluation on various datasets as well as different models. The comparison with other baselines demonstrates the effectiveness of the proposed algorithm.


Weaknesses:
1. The contribution of the proposed algorithm mainly lies in the optimal quantization range. However, the optimization of quantization range is not a new story in this field. Although the authors demonstrate some trial empirical analysis of the connection between reduced quantization range and bit error rate, these trial experiments might not be sufficient and convincing to make the conclusion that a reduced quantization range leads to better performance. Specifically, the empirical evidence excludes popular backbones and benchmarks. Second, there is no evidence that. Furthermore, it is difficult to see the influence of how $\alpha^*$ changes in layers with different depths.
2. The authors discuss the training with noise injection, which is a common practice in adversarial robustness, such as adversarial training. In Eq. 8, $\epsilon$ seems to be randomly sampled instead of adversarial attacks. More evaluation of the proposed algorithm with adversarial training is not included.

---

> ### Author Response · Authors · 2023-09-24
>
> We appreciate the reviewer for taking the time to provide feedback and insightful comments. We will now address the questions raised:
>
> > The contribution of the proposed algorithm mainly lies in the optimal quantization range. However, the optimization of quantization range is not a new story in this field. Although the authors demonstrate some trial empirical analysis of the connection between reduced quantization range and bit error rate, these trial experiments might not be sufficient and convincing to make the conclusion that a reduced quantization range leads to better performance.
>
> We acknowledge the reviewer's concern regarding the contribution of our approach. However, we believe that our paper empirically demonstrates the crucial role of quantization range in mitigating bit-flip noise across a wide range of models and datasets, which has not been extensively explored in prior research.
>
> > Furthermore, it is difficult to see the influence of how alpha_star changes in layers with different depths.
>
> As discussed in Section 3.5, we heuristically select the alpha_star for each layer to minimize the MSE between the dequantized weights and real weights. This selection process depends on both the weight distribution and the size of the weight, which can vary between layers with different depths.
>
> > Please justify the novelty.
>
> We understand the reviewer's request for justification of novelty.  In response to this concern, we have enhanced the manuscript by presenting additional empirical evidence and analysis to highlight the effectiveness of our method. We have also modified Section 2.3 to provide a clearer distinction between our proposed method and existing ones, emphasizing the motivation and novelty of our approach.
>
> > Please clarify the motivation and provide more empirical evidence and analysis.
>
> We have addressed this concern by including adversarial training as a new baseline to compare our method against bit-flip attacks. Additionally, we have introduced stealthy targeted attacks as a new benchmark and compared our method with OCM, another baseline, against these attacks. For further details and discussion, please refer to Appendix F. We believe that these modifications provide a more comprehensive evaluation of our algorithm's robustness.
>
> > Please provide more evaluation with adversarial training against adversarial attacks.
>
> We have included adversarial training in Appendix E and provided detailed discussions on the results. For a more comprehensive assessment, please refer to Appendix E.
>
> > Although this paper focuses on bit-flip noise, it could be interesting to see whether the proposed algorithm performs better robustness under traditional adversarial attacks on the image input, such as FGSM, PGD, etc.
>
> While our primary focus is on enhancing robustness against weight perturbations, we acknowledge the importance of evaluating our proposed algorithm's performance against traditional adversarial attacks like FGSM and PGD on the input data. In future work, we plan to include experiments to investigate the WCAT's robustness in these scenarios. However, we want to emphasize that our primary contribution lies in addressing robustness against weight perturbations.
>
> We truly appreciate the reviewer's feedback, and we hope these clarifications and updates have addressed their concerns adequately.

---

### Review · Reviewer_B8pi · 2023-09-08

**Summary Of Contributions:**

This paper focuses on the problem of improving robustness in quantized DNNs against random or adversarial bit-flips. Authors propose a weight clipping-aware training (WCAT) approach, where the quantization range of weights in each layer are independently optimized during training. By using a regularized training scheme, this learnable quantization range can be penalized/reduced in magnitude, which in turn increases robustness. In this pipeline, additional bit-flip noise is also incorporated during training to boost robustness further. Experiments are performed on CIFAR and ImageNet datasets, including ResNet and DeiT architectures at 4-bit and 8-bit quantization.

**Audience:**

Yes

**Broader Impact Concerns:**

I do not see any broader impact concerns.

**Claims And Evidence:**

Yes

**Requested Changes:**

(1) Setting a non-zero bit error rate $p_o$ during training seems beneficial against random bit-flips, which is intuitive and accurate. Is there any reliable evidence that it is also beneficial against adversarial bit-flip attacks? Can the authors elaborate Figures 5 & 6 in terms of the results without BFN in WCAT?

(2) Regarding SOTA comparison: Literature coverage of defenses is only centered around weight clipping and bit-flip based training methods. Some of the other existing adversarial bit-flip defenses are missing, and these were also found powerful in non-adversarial settings [1,2]. It would be good to include these methods for comparisons in the paper.

(3) Did/can the authors simulate any targeted bit-flip attack [3] scenarios in at least CIFAR level experiments? There is a significant line of attack/defense works exclusively focusing on targeted bit-flips [2-5], which constitute a highly realistic and important scenario. Authors should also discuss these relevant works for a broader inclusion of literature.

(4) Adversarial BFA setting often converges to solutions (bits to be flipped) in the final dense layer of the network. Did the authors simulate the "random attacks" considering the whole network or only the final dense layer(s)? Both would be interesting to see comparatively on e.g., one dataset.

Minor comments:
- Since bit-flip noise is a part of Algorithm 1, one assumes that it is already a part of WCAT by default. However in Figure 2 legend, there is a WCAT vs. WCAT+BFN separation. Perhaps one should restructure it in terms of setting $p_o=0$ or not.
- In Appendix B.4, for the statement: ".. we selected 64 sample images from the test set [...] average over 10 different seeds." -> I believe this means: ".. we sampled 64 images from the test set [...] average over 10 different seeds" ?
- In Figure 6, am I correct to assume that the title texts "(5M)", "(22M)" and "(86M)" indicate the number of parameters but not the # bits that can be flipped? I guess the latter would also be fitting here.

References:

[1] "Defending and harnessing the bit-flip based adversarial weight attack", CVPR 2020.

[2] "Improving robustness against stealthy weight bit-flip attacks by output code matching", CVPR 2022.

[3] "T-BFA: Targeted bit-flip adversarial weight attack", IEEE TPAMI 2021.

[4] "Targeted attack against deep neural networks via flipping limited weight bits", ICLR 2021.

[5] "Aegis: Mitigating Targeted Bit-flip Attacks against Deep Neural Networks",  USENIX Security Symposium 2023.

**Strengths And Weaknesses:**

Strengths:
- A novel solution to the niche problem of post-deployment bit-flip robustness of DNNs.
- Storyline of the paper is well constructed, the manuscript is very clear to follow, and no misleading/unsupported claims present.
- Paper does a good job at demonstrating the effectiveness of WCAT, including transformer-based models and ImageNet scale simulations. Design choice justifications (e.g., quantization range initialization, facilitating backpropagation) are described well.

Weaknesses:
- Mainly significant results are based on random bit-flip attacks. Adversarial attack experiments (Sec 4.6) can be extended to include targeted settings.
- Authors could improve their SOTA discussions on the relevant literature beyond some selected pieces of work.

---

> ### Author Response · Authors · 2023-09-24
>
> We thank the reviewer for their input and helpful comments. Now we address the raised questions and concerns.
>
> > (1) Setting a non-zero bit error rate <something goes here!> during training seems beneficial against random bit-flips, which is intuitive and accurate. Is there any reliable evidence that it is also beneficial against adversarial bit-flip attacks?  Can the authors elaborate Figures 5 & 6 in terms of the results without BFN in WCAT?
>
> In Figures 5 and 6, the models are selected among the best configurations for each quantization method. It's important to note that in Appendix B.4, we explained why we don't use BFN for ImageNet results. For normal quantization, WCAT, and LSQ, BFN improved robustness against adversarial attacks, and we used p_{o} = 0.5% for each of them. To individually show the effect of training using BFN in robustness against adversarial attacks, we added a new Table 3 in Appendix E. As we can see, the improvement for normal quantization and WCAT is slight. Also, for PLClip, we couldn't observe any improvement.
>
> > (2) Regarding SOTA comparison: Literature coverage of defenses is only centered around weight clipping and bit-flip based training methods. Some of the other existing adversarial bit-flip defenses are missing, and these were also found powerful in non-adversarial settings [1,2]. It would be good to include these methods for comparisons in the paper.
>
> We appreciate the reviewer's suggestion for a broader discussion of relevant literature. Reference [1] proposed binarization as a robust quantization method against adversarial attacks. However, as we outlined in Section 2.3, while binarization indeed enhances robustness, it does so at the expense of clean performance. Additionally, the scalability of this method, especially on transformers, is a concern. In response to this feedback, we have included Reference [2] as a new baseline and compared our method with it on targeted attacks. Please refer to Appendix F for further details.
>
> > (3) Did/can the authors simulate any targeted bit-flip attack [3] scenarios in at least CIFAR level experiments? There is a significant line of attack/defense works exclusively focusing on targeted bit-flips [2-5], which constitute a highly realistic and important scenario. Authors should also discuss these relevant works for a broader inclusion of literature.
>
> Following the reviewer's advice, we have included the stealthy targeted bit-flip attack [3] as a new benchmark in Appendix F. For more discussion, please refer to Appendix F.
>
> > (4) Adversarial BFA setting often converges to solutions (bits to be flipped) in the final dense layer of the network. Did the authors simulate the "random attacks" considering the whole network or only the final dense layer(s)? Both would be interesting to see comparatively on e.g., one dataset.
>
> We simulated the random attack by injecting bit error noise into all convolutional and fully connected layers, including the final dense layer. For adversarial attacks, we added Figures 14 and 15 to show the distribution of layers affected by BFA and stealthy T-BFA. We observed that for stealthy T-BFA, the bit-flip noise mainly impacts the last two layers, but for BFA, the first convolutional layer, as well as intermediate layers, are predominantly targeted. For more discussion, please refer to Appendix F.
>
> > Since bit-flip noise is a part of Algorithm 1, one assumes that it is already a part of WCAT by default. However in Figure 2 legend, there is a WCAT vs. WCAT+BFN separation. Perhaps one should restructure it in terms of setting or not.
>
> We have updated the legend of the mentioned figure to avoid any confusion. Please find the updated version in Figure 1.
>
> > In Appendix B.4, for the statement: ".. we selected 64 sample images from the test set [...] average over 10 different seeds." -> I believe this means: ".. we sampled 64 images from the test set [...] average over 10 different seeds" ?
>
> We have edited Appendix B.4 to clarify that we randomly sample 64 images from the test set as the default configuration. Subsequently, we calculate the test accuracy using the remaining data and report the average value of N_{flip} across 10 different seed values.
>
> > In Figure 6, am I correct to assume that the title texts "(5M)", "(22M)" and "(86M)" indicate the number of parameters but not the # bits that can be flipped? I guess the latter would also be fitting here.
>
> You are correct. The titles "(5M)", "(22M)", and "(86M)" indicate the number of parameters, not the number of bits that can be flipped. To avoid any confusion, we have removed these indications. As bit-flip noise is injected into all convolutional and fully connected layers, most of the DeiT parameters belong to these layers, and we wanted to show the effect of scaling on robustness for all quantization methods.
>
> We're grateful for the reviewer's feedback and time. We hope our explanations have resolved any concerns.

---

> > ### Comment · Reviewer_B8pi · 2023-09-26
> > **response to the authors**
> >
> > Thanks to the authors for their detailed responses and changes to the manuscript.
> >
> > Several clarifications were made, and all of my concerns are addressed. Importantly, the lack of experimental focus on adversarial BFA experiments was revisited. Additional simulations on targeted bit-flip attacks are quite thorough. I am happy with the current version of the manuscript, and I recommend acceptance.
> >
> > One minor comment on removing the titles "(5M)", "(22M)", "(86M)": They are removed from Fig 5, but in Fig. 3 they are still present. Please correct.

---

### Review · Reviewer_NCJM · 2023-09-09

**Summary Of Contributions:**

This paper proposes a weight clipping-aware training (WCAT) framework to improve the robustness of quantized neural networks against bit-flip weight perturbations. The motivation behind WCAT is an empirical observation on the relationship between quantization range and robustness in a toy model. The critical part of implementing WCAT is optimizing the quantization range during training. Furthermore, WCAT can be combined with the Bit-flip noise (BFN) injection method to further improve robustness. Experiments show that WCAT can outperform some baseline methods in terms of robustness against bit errors.

**Audience:**

Yes

**Claims And Evidence:**

Yes

**Requested Changes:**

1. Provide a clearer motivation for this paper. This can be achieved by:
    - Explaining why learning the quantization range with optimization is better than existing methods, or
    - Justifying how the motivation discussed in Appendix A is related to the proposed method.
2. Add experiments with more state-of-the-art baselines, as stated above.
3. Reorganize Section 3:
    - I suggest delaying the presentation of Algorithm 1 to the end of Section 3. For instance, you can add a subsection to summarize the WCAT framework by going through Algorithm 1. By contrast, putting Algorithm 1 at the start makes it somewhat hard to digest.
    - The experiment provided in Figure 1 should appear in Section 4 along with other ablation studies, since this section aims to introduce the method.
4. Reorganize Section 4:
    - I suggest putting the major comparison (robustness) in the front, and other ablation studies in the latter.
    - Some comparisons like Figure 2 may be better illustrated in tables, as the results seem to be very close.

**Strengths And Weaknesses:**

**Strengths**

1. The proposed method is clear and easy to follow.
2. Each component of WCAT, including range initialization and regularization, is discussed and analyzed in detail through experiments.
3. The experiment covers different model architectures, attacks, and datasets, which sufficiently support some of the claims made in this paper.

**Weaknesses**

1. Using per-layer weight clipping is already proposed in [1], which limits the novelty of this paper.
2. The motivation provided in Appendix A is confusing. To the best of my understanding, this observation supports the claim that "increasing the bit error rate leads to a smaller optimal quantization range." However, WCAT proposes to **optimize** the quantization range for different layers. In my opinion, the major difference between this work and existing methods like [1, 2] is using optimization to learn the quantization range. Therefore, the motivation should explain why optimization is better than the specific quantization range proposed in [1].
3. The baselines seem to be incomplete in the experiment settings. The main comparison includes WCAT, PLClip, and LSQ, but there are other methods like AdvBET and RandBET proposed in [1]. Given that [1] may be the most related work to this paper, I consider comparison to [1] is crucial.
4. The organization of Sections 3 and 4 may be somewhat confusing, which affects the readability of this paper. Please refer to "Requested Changes" for more details.

[1] Random and adversarial bit error robustness: Energy-efficient and secure DNN accelerators. *IEEE Transactions on Pattern Analysis and Machine Intelligence*, 2022.

[2] Bit error robustness for energy-efficient DNN accelerators. *Machine Learning and Systems*, 2021.

---

> ### Author Response · Authors · 2023-09-24
>
> We thank the reviewer for their time in reviewing our draft and providing useful feedback. We now address the concerns.
>
>
> > The motivation provided in Appendix A is confusing. To the best of my understanding, this observation supports the claim that "increasing the bit error rate leads to a smaller optimal quantization range." However, WCAT proposes to optimize the quantization range for different layers. In my opinion, the major difference between this work and existing methods like [1, 2] is using optimization to learn the quantization range. Therefore, the motivation should explain why optimization is better than the specific quantization range proposed in [1].
>
> We acknowledge the reviewer's request for a clearer motivation for our paper. As we discussed the aim of the proposed method is to achieve a balance between the robustness and noiseless performance of the model.  In Appendix A, we attempt to demonstrate the importance of the quantization range's impact on the robustness of the model using a toy example. We show that reducing the quantization range leads to better robustness, But on the other hand, it also decreases the dynamic range, which is vital in neural networks for representing weights with large absolute values [1]. So, the motivation for learning the quantization range through optimization is that we can dynamically adjust quantization ranges during training to strike a balance between achieving better performance and improved robustness. Also, it has been empirically demonstrated that learning quantization parameters can significantly improve the performance of quantized models.
>
> > The baselines seem to be incomplete in the experiment settings. The main comparison includes WCAT, PLClip, and LSQ, but there are other methods like AdvBET and RandBET proposed in [1]. Given that [1] may be the most related work to this paper, I consider comparison to [1] is crucial.
>
> AdvBET and RandBET are built upon PLClip, and the Bit-flip noise (BFN) injection method we propose in Section 3.4 is similar to RandBET, as both involve injecting random noise during training. As the reviewer rightly points out the importance of evaluating our algorithm with adversarial training, we have included it as a new baseline in our revised manuscript. For further details, please refer to Appendix E.
>
> > Provide a clearer motivation for this paper. This can be achieved by:
> Explaining why learning the quantization range with optimization is better than existing methods, or
> Justifying how the motivation discussed in Appendix A is related to the proposed method.
>
> We have modified Section 2.3 to first explain the differences between our proposed method and PLCLip, and then delve into the motivation behind learning the quantization range. Please refer to this section for further details.
>
>
> > Add experiments with more state-of-the-art baselines, as stated above.
>
> we have included adversarial training as a baseline for comparison with our proposed method in Appendix E. Additionally, we have introduced targeted adversarial attacks [2] as a new benchmark and compared our method with OCM [3] as another new baseline, considering it a state-of-the-art approach for robustness against stealthy targeted attacks. Please refer to Appendix F for further discussion.
>
> > Reorganize Section 3:
> I suggest delaying the presentation of Algorithm 1 to the end of Section 3. For instance, you can add a subsection to summarize the WCAT framework by going through Algorithm 1. By contrast, putting Algorithm 1 at the start makes it somewhat hard to digest.
> The experiment provided in Figure 1 should appear in Section 4 along with other ablation studies, since this section aims to introduce the method.
>
> We agree with the reviewer's suggestion to improve the readability of Section 3. Accordingly, we have restructured the section by moving Algorithm 1 to the end and adding subsection 3.6 to provide a summarized overview of the WCAT framework.
>
> > Reorganize Section 4:
> I suggest putting the major comparison (robustness) in the front, and other ablation studies in the latter.
> Some comparisons like Figure 2 may be better illustrated in tables, as the results seem to be very close.
>
> Following the reviewer's advice, we have reorganized Section 4 by placing the major comparison at the front and positioning the ablation studies toward the end and we moved Figure 1 in Section 4.6 along with other ablation studies.
>
> We truly value the reviewer's feedback, and we sincerely hope that our clarifications have addressed their concerns.
>
>
> References:
>
> [1] Random and adversarial bit error robustness: Energy-efficient and secure DNN accelerators. IEEE Transactions on Pattern Analysis and Machine Intelligence, 2022.
>
> [2] "T-BFA: Targeted bit-flip adversarial weight attack", IEEE TPAMI 2021.
>
> [3] "Improving robustness against stealthy weight bit-flip attacks by output code matching", CVPR 2022.

---

> > ### Comment · Reviewer_NCJM · 2023-09-24
> > **Thank you for the response**
> >
> > I thank the authors for their response. After reading their rebuttal and revision, I'm satisfied with the current version of the paper, where the motivation becomes more clear and more comprehensive evaluations are involved. Overall, I recommend accepting this paper.

---

### Author Response · Authors · 2023-09-24

We sincerely appreciate the reviewers for their valuable feedback and insightful comments on our paper. We are glad that the reviewers found our paper to be clearly written and easy to follow. We are pleased that reviewers think we show enough evidence to support our claims (Reviewer **B8pi** and **eJU5**), and would like to particularly express our gratitude to Reviewer **B8pi** for highlighting that our paper contains “no misleading/unsupported claims”. We are also grateful to see that we perform an “extensive evaluation” (Reviewer eJU5) that shows “the effectiveness of the proposed algorithm” (Reviewer **eJU5** and **B8pi**), and that our design choices are “discussed and analyzed in detail” (Reviewer **NCJM**) and “described well (Reviewer **B8pi**)”.

We would like to address the common concerns raised by the reviewers below and provide individual responses to specific questions in their reviews. All reviewers emphasized the importance of evaluating our algorithm with adversarial training against adversarial attacks. We acknowledge this critical point, and have incorporated such evaluations into our revised manuscript in Appendix E to provide a more comprehensive assessment of the robustness of our approach. All the changes in the revised manuscript are in blue text. Additionally, the section names in blue indicate that they have been reordered.

As presented in Appendix E, we first observed that adversarial training indeed enhances the model's robustness against BFA when using normal quantization. However, when compared to other robust quantization methods, it remains significantly more susceptible to adversarial attacks while also compromising clean performance. Additionally, it is crucial to note that adversarial training significantly extends the training time due to the repeated execution of the BFA at each training iteration. We kindly ask the reviewers to refer to Appendix E for a more detailed discussion.

---

### Decision · Action_Editor_JrEy · 2023-10-17

**Recommendation:** Accept as is

**Comment:**

This paper proposes a weight clipping-aware training (WCAT) approach to enhance DNN robustness against bit-flip errors on model weights. WCAT helps to achieve an optimized dynamic range of quantized weights in each layer to strike a balance between model accuracy and robustness. The effectiveness of the methods is evaluated on ResNet and DeiT models and on CIFAR and ImageNet datasets under a range of robustness benchmarking settings.

The paper is well written. Although the introduced quantization technique, i.e. the trainable clip values has been well-established in prior research on model quantization, the authors re-directed their attention toward improving the robust. The results clearly illustrate the efficacy of the methods in achieving a balance between model accuracy and robustness.

Reviewers have raised two main concerns:

1). The motivation is not clearly presented.

2). Lack evaluation on adversarial training and targeted bit-flip attack.

During the rebuttal phase, the authors thoroughly address the concerns and provide additional data.

In summary, this paper is technically solid. Two reviewers recommend accept, while one reviewer leaning rejection concerning the limited backbone structures evaluated in the paper. However, considering the proposed methods demonstrate competitive results and offer potential benefits in this domain, I recommend acceptance. I also hope that the authors will conduct a wider range of evaluations in their future work.

**Audience:**

Yes. This work present techniques and empirical evidence that make progress in improving the resilience of DNNs against bit-flip errors.

**Claims And Evidence:**

The claims are substantiated with clear evidence, particularly, after the authors provided additional evaluations on adversarial training and targeted bit-flip attack during rebuttal.